# Pretreatment with a Heat-Killed Probiotic Modulates the NLRP3 Inflammasome and Attenuates Colitis-Associated Colorectal Cancer in Mice

**DOI:** 10.3390/nu11030516

**Published:** 2019-02-28

**Authors:** I-Che Chung, Chun-Nan OuYang, Sheng-Ning Yuan, Hsin-Chung Lin, Kuo-Yang Huang, Pao-Shu Wu, Chia-Yuan Liu, Kuen-Jou Tsai, Lai-Keng Loi, Yu-Jen Chen, An-Ko Chung, David M. Ojcius, Yu-Sun Chang, Lih-Chyang Chen

**Affiliations:** 1Molecular Medicine Research Center, Chang Gung University, Taoyuan 333, Taiwan; ycc0311@gmail.com (I.-C.C.); oychunnan@gmail.com (C.-N.O.); ishucab@gmail.com (S.-N.Y.); 2Graduate Institute of Medical Sciences, National Defense Medical Center, Taipei 114, Taiwan; hsinchunglin@gmail.com; 3Division of Clinical Pathology, Department of Pathology, Tri-Service General Hospital, Taipei 114, Taiwan; 4Graduate Institute of Pathology and Parasitology, National Defense Medical Center, Taipei 114, Taiwan; cguhgy6934@gmail.com; 5Department of Pathology, Mackay Memorial Hospital, New Taipei City 251, Taiwan; pw2136@gmail.com; 6Department of Medicine, Mackay Medical College, New Taipei City 252, Taiwan; t109@mmc.edu.tw; 7Division of Gastroenterology, Department of Internal Medicine, MacKay Memorial Hospital, New Taipei City 251, Taiwan; 8Department of Medical Research, MacKay Memorial Hospital, New Taipei City 251, Taiwan; chenmdphd@gmail.com; 9Department of Laboratory Medicine, MacKay Memorial Hospital, Taipei 104, Taiwan; benson@mmh.org.tw; 10Department of Dentistry, School of Dentistry, National Yang-Minutesg University, Taipei 112, Taiwan; keng1245@gmail.com; 11Department of Radiation Oncology, Mackay Memorial Hospital, New Taipei City 251, Taiwan; 12Graduate Institute of Biomedical Sciences, College of Medicine, Chang Gung University, Taoyuan 333, Taiwan; sevemthday@hotmail.com; 13Department of Biomedical Sciences, University of the Pacific Arthur A. Dugoni School of Dentistry, San Francisco, CA 94103, USA; dojcius@pacific.edu; 14Center for Molecular and Clinical Immunology, Chang Gung University, Taoyuan 333, Taiwan; 15Chang Gung Immunology Consortium, Chang Gung Memorial Hospital, Linkou 333, Taiwan; 16Department of Otolaryngology-Head & Neck Surgery, Chang Gung Memorial Hospital, Linkou 333, Taiwan

**Keywords:** probiotic, NLRP3, inflammasome, colitis, cancer

## Abstract

Colorectal cancer (CRC) is one of the most common malignancies worldwide. Inflammation contributes to cancer development and inflammatory bowel disease is an important risk factor for CRC. The aim of this study is to assess whether a widely used probiotic *Enterococcus faecalis* can modulate the NLRP3 inflammasome and protect against colitis and colitis-associated CRC. We studied the effect of heat-killed cells of *E. faecalis* on NLRP3 inflammasome activation in THP-1-derived macrophages. Pretreatment of *E. faecalis* or NLRP3 siRNA can inhibit NLRP3 inflammasome activation in macrophages in response to fecal content or commensal microbes, *P. mirabilis* or *E. coli*, according to the reduction of caspase-1 activation and IL-1β maturation. Mechanistically, *E. faecalis* attenuates the phagocytosis that is required for the full activation of the NLRP3 inflammasome. In in vivo mouse experiments, *E. faecalis* can ameliorate the severity of intestinal inflammation and thereby protect mice from dextran sodium sulfate (DSS)-induced colitis and the formation of CRC in wild type mice. On the other hand, *E. faecalis* cannot prevent DSS-induced colitis in NLRP3 knockout mice. Our findings indicate that application of the inactivated probiotic, *E. faecalis*, may be a useful and safe strategy for attenuation of NLRP3-mediated colitis and inflammation-associated colon carcinogenesis.

## 1. Introduction

Colorectal cancer (CRC) is one of the most common malignancies worldwide, with an estimated 135,430 new cases predicted for 2017 in the United States [1]. It is commonly accepted that inflammation contributes to cancer development, and inflammatory bowel disease (IBD) is known to be an important risk factor for CRC development [2]. IBD represents a group of intestinal disorders caused by inflammation of the gastrointestinal tract [3]. The two major types of IBD are ulcerative colitis and Crohn’s disease. Ulcerative colitis causes inflammation in the mucosa of the colon and rectum, while Crohn’s disease affects all parts of the gastrointestinal tract, from the mouth to the anus. Both diseases are characterized by a series of relapses and remissions. The etiology and pathogenesis of IBD remain largely unclear. Several lines of evidence suggest that dysbiosis in enteric microbiota and defective innate immunity are two key players [4].

*Enterococcus faecalis* is a Gram-positive and facultative anaerobic commensal bacterium that belongs to the group of lactic acid bacteria. For a long time now, *E. faecalis* has been widely used as a probiotic product [5]. The World Health Organization defines probiotics as “live microorganisms, which when adminutesistered in adequate amounts, confer a health benefit on the host”. However, mounting evidence indicates that dead lactic acid bacteria, including *E. faecalis*, may also exert immunomodulatory effects [6,7]. Studies have indicated that dead probiotics are safer than live probiotics due to the decreased risk for systematic infection and/or transfer of antibiotic genes [8]. Therefore, dead probiotics could be particularly beneficial in children and immunocompromised people [6,7]. The results of numerous clinical trials indicate that heat-killed probiotic bacteria can have health benefits, such as by reducing skin disease in adult atopic dermatitis patients [9], improving the life quality for patients with allergic rhinitis [10], decreasing the incidence of upper respiratory tract infection in healthy people with psychological stress [11], and enhancing immunity in elderly people [12]. Previous studies found that oral adminutesistration of lysed *E. faecalis* can reduce the allergen-induced immune response in mice [13] and that heat-killed *E. faecalis* can modulate monocyte chemoattractant protein-1 and reduce the pathogenicity of influenza and enterovirus 71 infections [7]. Notably, pretreatment with viable or heat-killed *E. faecalis* has been shown to have protective effects against colitis in dextran sodium sulfate (DSS)-induced colitis in mice and small intestinal cancer formation in Apc mutant Minutes mice [14,15]. However, the effect of *E. faecalis* pretreatment on colitis-associated CRC and the potential mechanisms underlying this protective effect remains largely unknown. 

NLRP3 inflammasomes are cytoplasmic multiprotein complexes that are important for innate immunity. They consist of the cytosolic pattern recognition receptor, NLRP3, the adaptor protein, ASC, and pro-caspase-1 [16]. The assembly of the NLRP3 inflammasome is responsible for activating pro-caspase 1 p45 to produce cleaved caspase-1 p10, which subsequently mediates the maturation of the pro-inflammatory cytokines, pro-IL-1β p31 and pro-IL-18 p24, to generate the secondsretable forms, IL-1β p17 and IL-18 p18. A two-signal model has been proposed to explain the regulation of the NLRP3 inflammasome [17]: the first signal (priminutesg) enables the expression of NLRP3, pro-IL-1β and pro-IL-18; and the secondsond signal (activation) is triggered by pathogen-associated molecule patterns (PAMPs) and damage-associated molecular patterns (DAMPs; e.g., nigericin [18] and ATP [19]) and leads to the assembly of the NLRP3 inflammasome. In colon tissues, NLRP3 and IL-1β are expressed in both healthy and colitic states [20]. The function of the NLRP3 inflammasome is important for healthy states, as seen in its contribution to the recovery of intestinal tissue damage in DSS-treated colitic mice [21]. However, excessive activation of the NLRP3 inflammasome results in development of several inflammatory diseases, including septic shock [22], type 2 diabetes [23], cryopyrin-associated periodic syndromes [24], rheumatoid arthritis [25], and Alzheimer’s disease [26]. Recently, Seo et al. provided evidence that activation of the NLRP3 inflammasome is also involved in promoting colitis in the DSS-treated mouse model [27]. Finally, studies have shown that certain members of the microbiota, especially *Proteus mirabilis*, can induce robust IL-1β production via the NLRP3 inflammasome in recruited inflammatory macrophages, and thereby promote IL-1β-dependent inflammation and damage in the intestine [27].

Here, we demonstrate that pretreatment with heat-killed *E. faecalis* can inhibit the induction of IL-1β secondsretion in macrophages stimulated with fecal content and two commensal microbes, *P. mirabilis* and *E. coli*. Mechanistically, *E. faecalis* attenuates the phagocytosis that is required for full activation of the NLRP3 inflammasome. Finally, we show that *E. faecalis* can ameliorate the severity of intestinal inflammation and protect mice from DSS-induced colitis and the formation of CRC.

## 2. Materials and Methods 

### 2.1. Reagents and Antibodies

PMA (cat# P1585), ATP (cat# A7699), nigericin (cat# N7143), 4′,6-diamidino-2-phenylindole (DAPI; cat# D9542), and cytochalasin D (cat# C8273) were purchased from Sigma-Aldrich (St Louis, MO, USA), anti-ASC (cat# SC-22514-R), anti-human caspase-1 (cat# SC-56036), anti-mouse caspase-1 (cat# SC-514), anti-human IL-1β (cat# SC-32294), anti-α-tubulin (cat# SC-32293), anti-GAPDH (cat# SC-32233), and goat anti-rabbit IgG-horseradish peroxidase (HRP; cat# SC-2004) from Santa Cruz Biotechnology (Santa Cruz, CA, USA); anti-NLRP3 (cat# AG-20B-0014) from Adipogen (San Diego, CA, USA), anti-mouse IL-1β (cat# AF-401-NA) from R&D Systems Inc. (Minutesneapolis, MN, USA); sheep anti-mouse IgG-HRP (cat# NA931) from Amersham (Amersham, UK), (5-(and-6)-carboxyfluorescein diacetate, succinimidyl ester (CFSE; cat# C1157) and Alexa Fluor-594 conjugated goat-anti-mouse IgG (H+L) (cat# A-11005) from Invitrogen (Carlsbad, CA, USA); and fluoresbrite yellow green carboxylate microspheres (1-μm YG beads, cat# 15702) from Polysciences Inc. (Warrington, PA, USA).

### 2.2. Preparation of Probiotic, Bacteria, and Fecal Content

The probiotic *E. faecalis* strain KH2 (Cosmo Foods, Tokyo, Japan) was originally isolated from a fruit. *E. faecalis* cells were killed by heat treatment at 80 °C for 30 min. The heat-killed *E. faecalis* were lyophilized and stored at −80 °C until use. *E. coli* (DH5α) was obtained from Real Biotech Corporation. *P. mirabilis* (ATCC 12453) was obtained from the American Type Culture Collection. For preparation of fecal content, fresh fecal pellets were collected from C57BL/6 mice, 100 mg/mL were vortexed in PBS for 30 min, and the fecal homogenate was collected by centrifugation at 1000 rpm for 5 s. For heat inactivation, the fecal content was heat treated at 100 °C for 5 min. For UV inactivation, the fecal content was exposed to UV for 30 min. The number of CFU per milliliter of bacterial suspension was calculated using a DensiCHEK instrument (bioMerieux Inc., Hazelwood, MO, USA).

### 2.3. Cell Culture

The human leukemia monocytic THP-1 cell line was maintained in RPMI and stimulated to macrophages with 200 nM PMA for 16 h as described previously [28]. For measurement of inflammasome activation, the THP-1 cells were treated with fecal content (1:200 dilution of stock), *E. coli* (MOI 100) or *P. mirabilis* (MOI 100) for 3 h, with 10 μM nigericin for 1 h, or with 5 mM ATP for 4 h. To examinutese the effects of *E. faecalis* on cells stimulated with fecal content, *E. coli* or *P. mirabilis*, PMA-differentiated THP-1 cells were pretreated with 1, 2, or 4 mg/mL of *E. faecalis* as indicated. At 24 h after *E. faecalis* incubation, THP-1 cells were stimulated with fecal content, heat-inactivated fecal content, UV-irradiated fecal content, *E. coli*, or *P. mirabilis* for 3 h.

### 2.4. RNA Interference

Transfection of the dsRNA duplexes (50 nM) was performed using Lipofectaminutese 2000 (Invitrogen) as previously described [29]. The cells were harvested and analyzed at two days post-transecondstion. The reagent used to target NLRP3 (cat# HSS132811-3, Invitrogen) included three 25-bp RNA duplexes: 5′-AAAGG AAGAA GACGU ACACC GCGGU-3′, 5′-ACCGC GGUGU ACGUC UUCUU CCUUU-3′ and 5′-UUAGC UUUGG CUUUC ACUUC AAUCC-3′.

### 2.5. Immunoblot Analysis

Immunoblot analysis has been described previously [28]. Briefly, cells were lysed in RIPA buffer containing a protease inhibitor cocktail on ice for 30 min. For measurement of IL-1β secondsretion and caspase-1 activation, culture supernatants were collected and concentrated by trichloroacetic acid (TCA; cat# T9159) precipitation. The protein lysates (from culture cells) or precipitated protein samples (from culture supernatants) were resolved by SDS-PAGE and transferred to nitrocellulose membranes (Amersham). The membranes were incubated with the indicated primary antibodies (1:1000 dilutions of indicated antibody except anti-GAPDH antibody, which was at 1:2000), and then with an HRP-conjugated secondsondary antibody (1:5000 dilutions of each antibody). The immunoreactive bands were detected using the TOOLS Extreme ECL-HRP Substrate (cat# TU-ECL03; Biotools, Taiwan).

### 2.6. CFSE Labeling of Bacteria and Fecal Content

Bacteria and fecal content were washed twice with PBS, suspended in PBS containing 10 μM CFSE, and incubated in for 30 min under gentle agitation at 37 °C. CFSE-labeled bacteria and fecal content were washed three times with PBS prior to use.

### 2.7. Phagocytosis Assay

PMA-differentiated THP-1 cells were preincubated with 2 mg/mL of *E. faecalis* for 24 h or with 10 μM cytochalasin D for 1 h, followed stimulation with YG beads (10 beads/cell), fecal content, *E. coli* (MOI 100) or *P. mirabilis* (MOI 100) for 3 h. Cells were stained for visualization of tubulin (1:100 dilution of anti-α-tubulin followed by a 1:1000 dilution of Alexa Fluor-594 conjugated secondsondary antibody (Invitrogen, red), YG beads, fecal content, *E. coli* or *P. mirabilis* (CFSE, green) and nuclei (DAPI, blue). To demonstrate the phagocytic efficiency of THP-1 cells, images were obtained with a confocal laser scanning microscope (LSM780; Carl Zeiss, Jena, Germany) and processed with the ZEN microscopic software (Carl Zeiss). The phagocytic activity was quantified by calculating the percentage of cells with internalized YG beads, CFSE-labeled fecal content or bacteria, as assessed using an IN cell analyzer (GE Healthcare, Freiburg, Germany).

### 2.8. ELISA

Cell culture supernatants were assayed for human IL-1β using a commercially available ELISA kit (cat# 88-7261; eBioscience, San Diego, CA, USA).

### 2.9. Animal Experiments

Mouse experiments were approved by and performed according to the guidelines of the Institutional Animal Care and User Committee of Chang-Gung University (CGU15-206). C57BL/6 mice were purchased from the National Laboratory Animal Center, Taiwan. Mice were maintained under specific pathogen-free conditions. For the induction of colitis, 2.5% DSS (molecular weight, 36,000–50,000; MP Biomedicals, Aurora, OH, USA) was given to mice (6–9 weeks old) via the drinking water as described previously [30]. To evaluate the effect of *E. faecalis* on DSS-induced colitis, mice were orally inoculated with *E. faecalis* (17 mg/kg) every day during the course of experiment, starting two weeks before the adminutesistration of DSS. For generation of colitis-associated CRC model, mice were injected intraperitoneally with azoxymethane (AOM; 10 mg/kg; cat# A5486; Sigma-Aldrich) prior to beginning the first of three or four cycles of DSS in drinking water as indicated. One cycle was defined as 6 days of DSS followed by 14 days of water. To evaluate the effect of *E. faecalis* on the mouse model of colitis-associated CRC, mice were orally inoculated with *E. faecalis* (17 mg/kg) every day during the course of experiment, starting two weeks before the adminutesistration of DSS or at the end of the third DSS treatment. Body weight and diarrhea were recorded daily. Scoring for severity of diarrhea was performed as described previously [31] with a slight modification. Briefly, diarrhea scores were determinutesed as follows: 0 points for well-formed stools, 1 point for soft but formed stools, 2 points for pasty and semiformed stools that did not adhere to the anus (mild diarrhea), 3 points for slimy stool (moderate diarrhea), 4 points for liquid stools that did adhere to the anus (severe diarrhea). The colon length and number of tumors per colon were measured at the time of sacrifice.

### 2.10. Statistical Analysis

Data from in vitro experiments and tumor growth in the mouse model were analyzed with the ANOVA test followed by Bonferroni post hoc test as indicated or Student’s *t*-test in SPSS as described previously [30]. Differences were considered significant at *p* < 0.05.

## 3. Results

### 3.1. NLRP3 Inflammasome is Required for Commensal Bacteria-Induced IL-1β Secondsretion

To confirm that IL-1β secondsretion can be induced through commensal bacteria-mediated NLRP3 inflammasome activation [27], we measured secondsreted IL-1β in cultures of THP-1-derived macrophages treated with fecal content or commensal bacteria. We assessed the expression of active caspase-1 p10 and mature IL-1β p17 by Western blot analysis, and examinutesed the level of IL-1β by ELISA. As shown in Figure 1a, we observed increased amounts of active caspase-1, mature IL-1β and secondsreted IL-1β in THP-1 cell cultures treated with fecal content, and found that these levels were reduced in cells treated with NLRP3-specific small interfering RNA (siRNA) but not control siRNA. Similarly, active caspase-1, mature IL-1β, and secondsreted IL-1β were induced by infection with either *E. coli* or *P*. *mirabilis*, and this induction was reduced by NLRP3 knockdown (Figure 1b). Our results show that fecal content and the tested commensal bacteria can induce IL-1β secondsretion through the NLRP3 inflammasome in macrophages.

### 3.2. Probiotic *E. faecalis* Attenuates Bacteria-Induced IL-1β Secondsretion

It has been reported that inactivated *E. faecalis* may improve allergy symptoms and reduce the pathogenicity of viral infection through immunomodulatory effects [7,13,32,33]. However, it was not known whether or how *E. faecalis* could modulate the commensal bacteria-induced activation of the NLRP3 inflammasome. We therefore evaluated the effect of *E. faecalis* pretreatment on inflammasome activation by assessing caspase-1 activation and IL-1β maturation in THP-1 cells. Our results indicated that the levels of active caspase-1 and mature IL-1β induced by either fecal content treatment (Figure 2a) or infection with *E. coli* or *P. mirabilis* (Figure 2b) were decreased in THP-1 cells pretreated with *E. faecalis* compared with untreated cells. Consistently, *E. faecalis* pretreatment also suppressed the IL-1β secondsretion induced by fecal content treatment (Figure 2c) or infection with *E. coli* or *P. mirabilis* (Figure 2d). The suppressive effect of *E. faecalis* on fecal content-induced IL-1β secondsretion appeared to be dose-dependent (Figure 2c), and the effect against commensal bacteria was observed in THP-1 cells infected with different doses of *E. coli* or *P. mirabilis* (Figure 2d). Taken together, our results suggest that *E. faecalis* pretreatment can efficiently attenuate fecal content- or commensal bacteria-induced NLRP3 inflammasome activation and IL-1β secondsretion. 

### 3.3. Probiotic *E. faecalis* Attenuates Bacteria-Induced NLRP3 Inflammasome Activation by Decreasing Bacterial Phagocytosis

Bacteria-induced NLRP3 inflammasome activation is thought to be regulated by at least two pathways [34,35]: in the first, phagocytosis of extracellular bacteria results in lysosomal destabilization and the release of cathepsin B; and in the secondsond, extracellular exposure to a bacterial pore-forminutesg toxin or ATP results in potassium efflux and a decrease of intracellular potassium. To explore which pathway is affected by *E. faecalis* treatment, we first examinutesed the effect of *E. faecalis* on phagocytosis in our system. The efficiencies of non-specific or bacterial phagocytosis were evaluated by monitoring the uptake of fluorescence-labeled polystyrene latex beads or bacteria, respectively. As shown in Figure 3a, pretreatment of *E. faecalis* caused clearly intracellular, phagocytosed *E. faecalis*. The uptake efficiency of fluorescent beads by THP-1 cells was significantly decreased when cells were pretreated with *E. faecalis* (Figure 3a). The phagocytosed *E. faecalis* appear to exhaust the capacity of phagocytosis in macrophages. Similarly, *E. faecalis* also significantly inhibited the phagocytosis of fecal content, *E. coli* and *P. mirabilis* (Figure 3b and Appendix A). Next, we used the phagocytosis inhibitor, cytochalasin D, to evaluate the role of phagocytosis in commensal bacteria-induced IL-1β secondsretion in our system. As shown in Figure 3c, cytochalasin D reduced the phagocytosis of fecal content and inhibited fecal content-induced IL-1β secondsretion (Figure 3d). Consistently, cytochalasin D also reduced *E. coli*- and *P. mirabilis*-induced IL-1β secondsretion (Figure 3d). These results indicate that bacterial phagocytosis is required for bacteria-induced NLRP3 inflammasome activation and IL-1β secondsretion, and that *E. faecalis* can interfere with phagocytosis to attenuate the bacteria-induced activation of the NLRP3 inflammasome.

### 3.4. Probiotic *E. faecalis* Induces Pro-IL-1β Expression but Does Not Affect ATP- or Nigericin-Induced NLRP3 Inflammasome Activation

Since bacteria can also activate the NLRP3 inflammasome through bacterial pore-forminutesg toxin- or ATP-mediated potassium efflux [34,35], we examinutesed the potential effect of *E. faecalis* on bacterial toxin- and ATP-induced NLRP3 inflammasome activation. As shown in Appendix A, *E. faecalis* significantly increased the ATP- or nigericin (a pore forminutesg toxin)-induced levels of mature IL-1β but did not affect the amount of active caspase-1. Consistent with increase of mature IL-1β, *E. faecalis* induced the expression of pro-IL-1β, regardless of the presence of ATP or nigericin (Appendix A). Furthermore, *E. faecalis* treatment increased IL-1β secondsretion in a dose-dependent manner, regardless of the presence of ATP or nigericin (Appendix A). Together, these results suggest that *E. faecalis* can contribute to ATP- and nigericin-induced IL-1β secondsretion through the upregulation of pro-IL-1β expression rather than through NLRP3 inflammasome activation.

### 3.5. *E. faecalis* Pretreatment Protects Mice from DSS-Induced Acute Colitis

Mounting evidence supports the idea that NLRP3 plays a pro-inflammatory role in colitis pathology [36,37,38,39]. In addition, it has been reported that blockage of IL-1β [27,40] or IL-18 [41] reduces colitis. As the NLRP3 inflammasome appears to enhance the development of colitis, it has been recommended as potential therapeutic target for the prevention of colitis and colitis-associated CRC [42]. Therefore, we assessed whether *E. faecalis* pretreatment can protect mice from acute colitis by reducing NLRP3 inflammasome activation, as assessed by monitoring weight loss and diarrhea in the DSS-treated mouse model (Figure 4a). Indeed, we found that *E. faecalis* significantly reduced DSS-induced weight loss compared to PBS control in wild type mice, but not in NLRP3-deficient mice (Figure 4b). Similarly, *E. faecalis*-treated mice had reduced diarrhea scores compared with PBS-treated control in wild type mice, but not in NLRP3-deficient mice (Figure 4c). Furthermore, the disease status was assessed by the colon length, which is used as an indicator of disease severity [30]. *E. faecalis*-treated mice had longer colons compared with PBS-treated control wild-type mice (8.5 cm vs. 7.6 cm), but no differences were observed in NLRP3-deficient mice (7.0 cm vs. 7.1 cm) (Figure 4d). Next, we examinutesed whether *E. faecalis* pretreatment affects inflammasome activation in vivo. As shown in Figure 4e, the levels of cleaved (activated) caspase-1 and mature IL-1β were reduced by *E. faecalis* in the colon tissues of mice with DSS-induced colitis compared to PBS control in wild type mice, but not in NLRP3-deficient mice. These results demonstrate that *E. faecalis* pretreatment can protect mice from DSS-induced acute colitis through an NLRP3-dependent manner.

### 3.6. *E. faecalis* Pretreatment Pprotects Mice from Colitis-Associated CRC

Since intestinal inflammation is an important risk factor for CRC development [2], we evaluated the ability of *E. faecalis* pretreatment to attenuate colitis-associated CRC. Mice were pretreated with *E. faecalis* and then treated with AOM and DSS to trigger the development of colitis-associated CRC (Figure 5a). *E. faecalis* pretreatment significantly ameliorated the weight loss of AOM/DSS-treated mice compared to PBS control mice (Figure 5b). *E. faecalis*-treated mice also presented with significantly lower diarrhea scores (Figure 5c) and had longer colons (8.0 cm vs. 7.3 cm) (Figure 5d,e) compared to PBS-treated control mice. Importantly, *E. faecalis* pretreatment inhibited the number of AOM/DSS-induced colon tumors per mouse (Figure 5e). We also evaluated the potential use of *E. faecalis* to treat the pre-existing CRC. Beginning at the end of the third DSS treatment, mice were inoculated with *E. faecalis* every day for 34 days (Appendix A). We did not observe any difference in % weight change (Appendix A) or diarrhea score (Appendix A) between *E. faecalis*-treated and PBS control mice. After daily inoculation with *E. faecalis* for 34 days, we also did not observe any difference in the colon length (Appendix A) or the number of colon tumors per mouse (Appendix Ae). Taken together, our results indicate that *E. faecalis* may exert protective effects against CRC but does not appear to act on pre-existing CRC. 

## 4. Discussion

Proper activation of the NLRP3 inflammasome is critical for immune defense, but improper activation causes inflammatory disease. Previous studies showed that the NLRP3 inflammasome contributes to the severity of DSS-induced acute colitis [36,37] and inherited spontaneous colitis [38,39]. Here, we introduce a useful and safe way to ameliorate acute colitis and attenuate colitis-associated CRC by oral treatment of heat-killed probiotic *E. faecalis*. Mechanistically, we show that pretreatment of *E. faecalis* significantly attenuates the fecal content- and intestinal commensal microbe-induced activation of the NLRP3 inflammasome in macrophages. Our results also reveal that *E. faecalis* interferes with phagocytosis, which is required for full activation of the NLRP3 inflammasome. Together, these findings provide functional and mechanistic evidence to support the use of *E. faecalis* for the attenuation of colitis and colitis-associated CRC. 

Bacterial environmental stimuli, conditioned by the inflammatory state of the intestine, are thought to be the primary trigger that causes a commensal microbe to transition into a pathogen [43]. Recently, Seo et al. provided evidence that certain members of the microbiota, especially *P. mirabilis*, can respond to the induction of DSS-induced colitis by inducing robust NLRP3 inflammasome-mediated IL-1β production in recruited inflammatory macrophages, when then promotes IL-1β-dependent inflammation and damage in the intestine [27]. Thus, appropriately limiting the inflammation induced by harmless commensal microbes could benefit people suffering from the intestinal inflammation, especially non-infectious inflammation, such as Crohn’s disease and chemotherapy-related diarrhea. Here, we show that *E. faecalis* pretreatment attenuates the NLRP3 inflammasome activation and IL-1β secondsretion in macrophages induced by fecal content or harmless commensal microbes, including *P. mirabilis* (Figure 2), and that this occurs through a decrease in phagocytosis (Figure 3). Importantly, *E. faecalis* pretreatment appeared to protect mice from DSS-induced acute colitis (Figure 4) and the development of colitis-associated CRC (Figure 5). This suggests that, consistent with the pro-inflammatory role of NLRP3 and IL-1β in intestinal inflammation, the *E. faecalis* pretreatment-induced attenuation of NLRP3 inflammasome activity in macrophages can ameliorate DSS-induced colitis. Our study supports the potential use of heat-killed *E. faecalis* to protect against intestinal inflammation initiated by non-infectious agents. Similar to our findings, the probiotic *E. coli* Nissle 1917 is markedly weaker than commensal *E. coli* K12 in its ability to activate the inflammasome [44] and also has the ability to ameliorate ulcerative colitis in humans [45] and mice [46]. However, the potential of *E. coli* Nissle 1917 to prevent colitis-associated colorectal cancer remains to be examinutesed. Beyond its effect on intestinal inflammation, the NLRP3 inflammasome is also involved in *Helicobacter pylori* infection-induced chronic gastric inflammation [47], which is the main cause of peptic ulcer disease and gastric cancer [48]. The degree of *H. pylori* infection-induced gastric inflammation was lower in NLRP3-deficient mice than wild type mice [47]. Therefore, it was proposed that the use of heat-killed *E. faecalis* may protect against peptic ulcer disease and gastric cancer in people infected with *H. pylori*.

In addition to being expressed in macrophages, NLRP3 inflammasome components are also expressed in intestinal epithelial cells, where they protect the mucosa by IL-18-dependently increasing the production of mucus by goblet cells [49]. Although *P. mirabilis* exists in the intestinal lumen and has contact with intestinal epithelial cells at the surface layer of the intestine, this commensal microbe usually does not provoke an inflammatory response, and keeps peace with the healthy host. Here, we observed that phagocytosis of this microbe is required for the ability of *P. mirabilis* to induce IL-1β secondsretion in macrophages (Figure 3). This suggests that *P. mirabilis* should efficiently activate the NLRP3 inflammasome in phagocytes, but not in non-phagocytic cells, such as intestinal epithelial cells. Thus, although *P. mirabilis* exists in the intestinal lumen, it should be restricted to the extracellular space and not enter the epithelial cells to directly activate the NLRP3 inflammasome under healthy conditions. When there is epithelial damage, such as following DSS treatment, *P. mirabilis* might cross the damaged epithelial layer from the intestinal lumen to laminutesa propria and reach the recruited inflammatory macrophages [43]. After undergoing phagocytosis, the engulfed *P. mirabilis* will strongly activate the NLRP3 inflammasome in inflammatory macrophages, leading to expansion of the intestinal inflammation. 

Considerable evidence supports the idea that NLRP3 inflammasome plays a pro-inflammatory role in colitis pathology [36,37,38,39], and blockage of IL-1β [27,40] or IL-18 [41] has been reported to reduce colitis. The NLRP3 inflammasome therefore appears to negatively affect the development of colitis. Indeed, it has been recommended as a potential therapeutic target for the prevention of colitis and colitis-associated CRC [42]. The efficacy of NLRP3 inhibitors has been examinutesed in experimental models of ulcerative colitis. For example, the diarylsulfonylurea-containing compound, MCC950, was found to inhibit the activation of the NLRP3 inflammasome, but not activation of the AIM2, NLRC4, or NLRP1 inflammasomes [50]. MCC950 can inhibit NLRP3 inflammasome activation and IL-1β secondsretion in response to *Escherichia coli* stimulation in macrophages [51]. In a spontaneous chronic colitis mouse model (Winnie mice), MCC950 reportedly attenuated colonic inflammation, as assessed by significant improvements in body weight and colon length, accompanied by reduced IL-1β secondsretion [39]. The benzo[d]imidazole derivative, Fc11a-2, was shown to inhibit the activation of the NLRP3 inflammasome and prevent the development of DSS-induced acute colitis [37]. However, these chemical inhibitors of the NLRP3 inflammasome may act directly on the NLRP3 molecule, and thus could inhibit NLRP3 inflammasome activity in both phagocytic and non-phagocytic cells. Here, we show that the probiotic *E. faecalis* interferes with phagocytosis to attenuate NLRP3 inflammasome activation (Figure 3), and does not act directly on the NLRP3 molecule (Appendix A). We therefore expect that *E. faecalis* does not affect the activation of the NLRP3 inflammasome in intestinal epithelial cells. Accordingly, pretreatment with *E. faecalis* should restrain the NLRP3 inflammasome activity in macrophages, which can worsen tissue damage in the inflamed intestine [27], without affecting NLRP3 activity in intestinal epithelial cells, which can aid in the recovery from intestinal tissue damage [49]. Of note, *E. faecalis* is present in the intestine and has been widely used as a probiotic microorganism in probiotic foods, whereas the precise mechanisms through which the chemical inhibitors, MCC950 or Fc11a-2, inhibit the NLRP3 inflammasome have not yet been clearly defined. Compared to MCC950 and Fc11a-2, the probiotic *E. faecalis* is expected to be safer. Thus, our present data collectively show a useful and safe way in which *E. faecalis* can be used to ameliorate colitis and colitis-associated CRC. 

Commensal microbes are usually harmless, but they may help promote intestinal inflammation when there is epithelial damage, such as in Crohn’s disease and chemotherapy-related diarrhea. Here, we provide evidence that pretreatment with heat-killed *E. faecalis* can efficiently attenuate NLRP3 inflammasome activation in macrophages, and that this occurs through negative effects on phagocytosis. Furthermore, *E. faecalis* can ameliorate the severity of DSS-induced murine experimental colitis and the development of colitis-associated CRC in mice. Together, our findings indicate that application of the probiotic, *E. faecalis*, may be a useful and safe strategy for treating colitis, especially that initiated by non-infectious agents.

## Figures and Tables

**Figure 1 nutrients-11-00516-f001:**
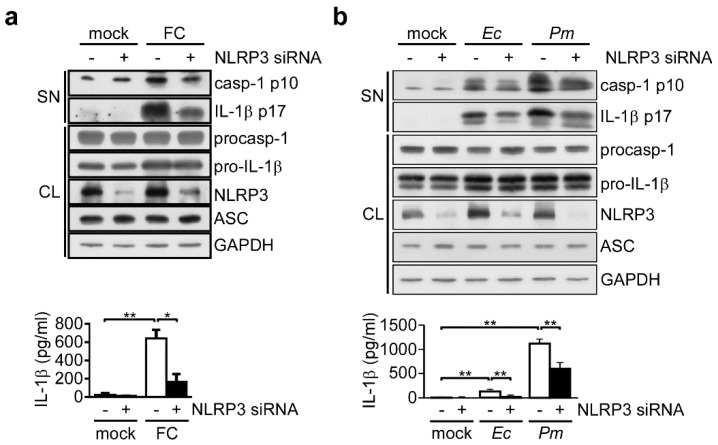
Requirement of the NLRP3 inflammasome for commensal bacteria-induced IL-1β secondsretion. (**a**,**b**) Top panels: Immunoblot analysis of NLRP3 inflammasome molecules in cell supernatants (SN) and cell lysates (CL). Bottom panels: ELISA of IL-1β in the SN of THP-1-derived macrophages treated with NLRP3 siRNA and then incubated for 3 h with (**a**) fecal content (FC), (**b**) *E. coli* (*Ec*) or *P*. *mirabilis* (*Pm*). The western blot is a representative of three independent experiments. ELISA data were analyzed with the ANOVA test followed by post hoc test. *, *p* < 0.05; and **, *p* < 0.01. All results are presented as the mean ± SD of three independent experiments. Abbreviations: casp-1 p10, active caspase-1 subunits; IL-1β p17, secondsreted mature IL-1β; procasp-1, p45 precursor of caspase-1; and pro-IL-1β, p31 precursor of IL-1β.

**Figure 2 nutrients-11-00516-f002:**
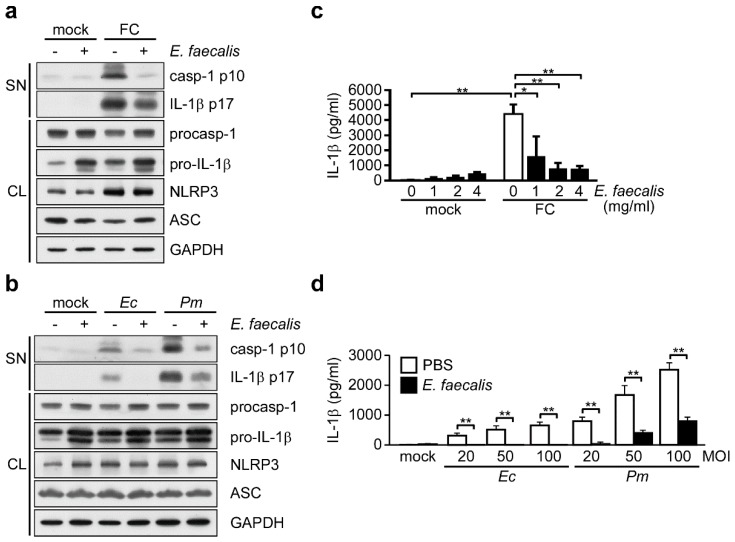
*E. faecalis* attenuates bacteria-induced IL-1β secondsretion. (**a**,**b**) Immunoblot analysis of NLRP3 inflammasome molecules in cell SN and CL from THP-1-derived macrophages pretreated with *E. faecalis* (2 mg/mL) for 24 h and then incubated for 3 h with (**a**) FC, (**b**) *Ec* (MOI 100) or *Pm* (MOI 100). The western blot is a representative of three independent experiments. (**c**,**d**) ELISA of IL-1β in the SN of THP-1-derived macrophages pretreated with *E. faecalis* at the dose as indicated (**c**) or 2 mg/mL (**d**) for 24 h and then infected for 3 h with FC (**c**), or *Ec* or *Pm* at 20, 50 or 100 MOI (**d**). ELISA data were analyzed with the ANOVA test followed by post hoc test. *, *p* < 0.05; and **, *p* < 0.01. All results are presented as the mean ± SD of three independent experiments. Abbreviations: casp-1 p10, active caspase-1 subunits; IL-1β p17, secondsreted mature IL-1β; procasp-1, p45 precursor of caspase-1; and pro-IL-1β, p31 precursor of IL-1β.

**Figure 3 nutrients-11-00516-f003:**
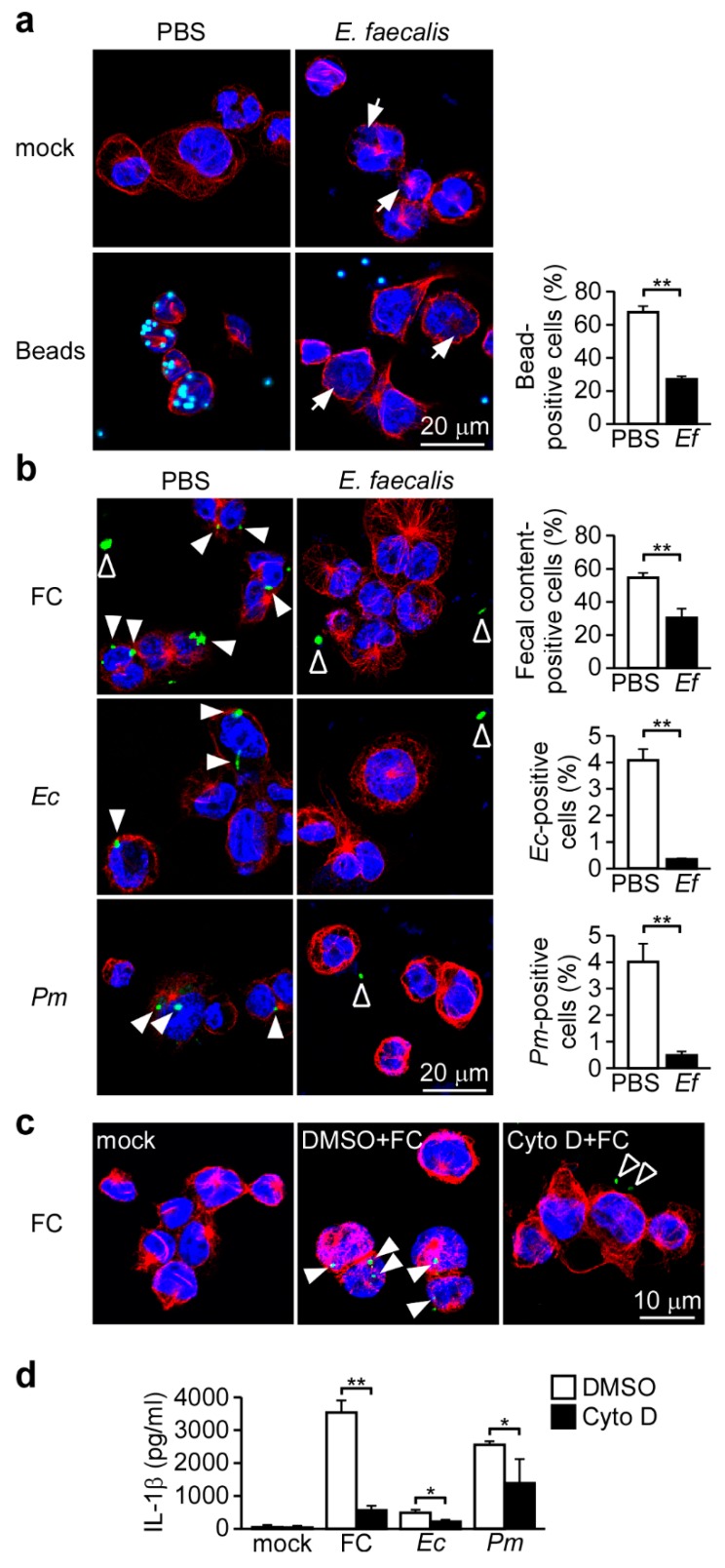
Phagocytosis, which is required for bacteria-induced inflammasome activation, is inhibited by *E. faecalis.* (**a**,**b**) THP-1-derived macrophages were pretreated with *E. faecalis* (Ef, 2 mg/mL) for 24 h, incubated for 3 h with (**a**) beads, (**b**) FC, Ec or *Pm*, and *assessed for phagocytosis*. Scale bars, 20 μm. Phagocytosis was quantified with an IN Cell Analyzer. (**c**) Assessment of phagocytosis in THP-1-derived macrophages pretreated with cytochalasin D for 1 h and then incubated for 3 h with FC. Scale bars, 10 μm. Cells were stained for visualization of tubulin (anti-α-tubulin, red), beads, FC, Ec or *Pm* (CFSE, green) and nuclei (DAPI, blue). (**d**) ELISA of IL-1β in the SN of THP-1-derived macrophages pretreated with cytochalasin D for 1 h and then infected for 3 h with FC, Ec or *Pm.* *, *p* < 0.05; and **, *p* < 0.01. Arrow, phagocytosed *E. faecalis*. Closed arrowhead, phagocytosed FC, Ec or *Pm*. Open arrowhead, unphagocytosed FC, Ec or *Pm*. These data are representative of three independent experiments.

**Figure 4 nutrients-11-00516-f004:**
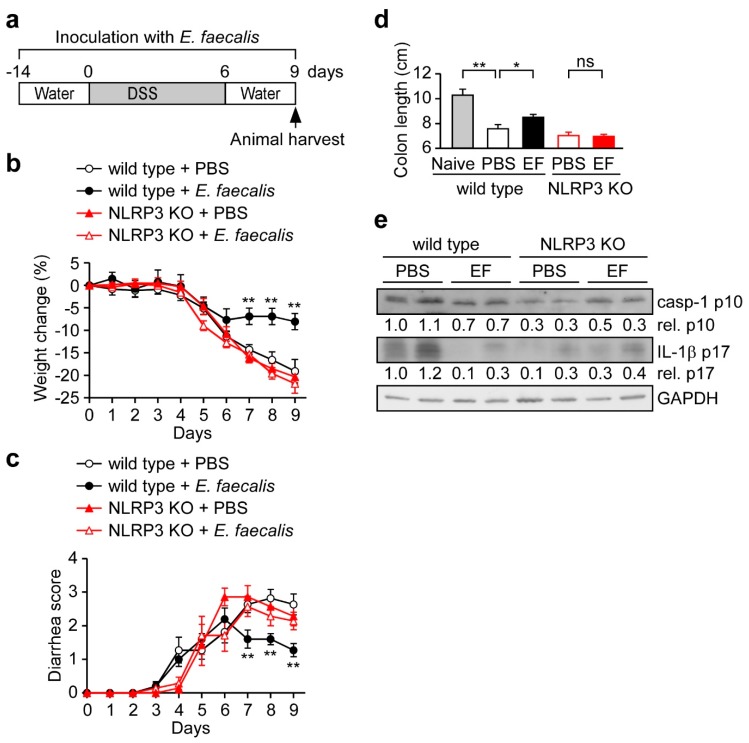
*E. faecalis* pretreatment protects mice from DSS-induced acute colitis. (**a**) Schematic presentation of the DSS-induced colitis model. Mice were orally inoculated with *E. faecalis* every day during the course of experiment, starting two weeks before the adminutesistration of DSS, which was given to mice via the drinking water for seven consecondsutive days to induce colitis. Nine days after the start of DSS treatment, all mice were sacrificed; *n* = 4 for the naïve control group, *n* = 11 for the PBS and *E. faecalis*-treated wild type mice group and *n* = 7 for the PBS and *E. faecalis*-treated NLRP3-deficient (KO) mice group. (**b**,**c**) The percent weight change (**b**) and diarrhea score (**c**) were monitored daily in all mice. (**d**,**e**) Colon length and colon tissue proteins were measured at the time *E. faecalis*-treated and PBS control mice were sacrificed. Proteins extracted from colon samples were immunoblotted using antibodies against caspase-1 p10, IL-1β p17, and GAPDH (**e**). *, *p* < 0.05; and **, *p* < 0.01.

**Figure 5 nutrients-11-00516-f005:**
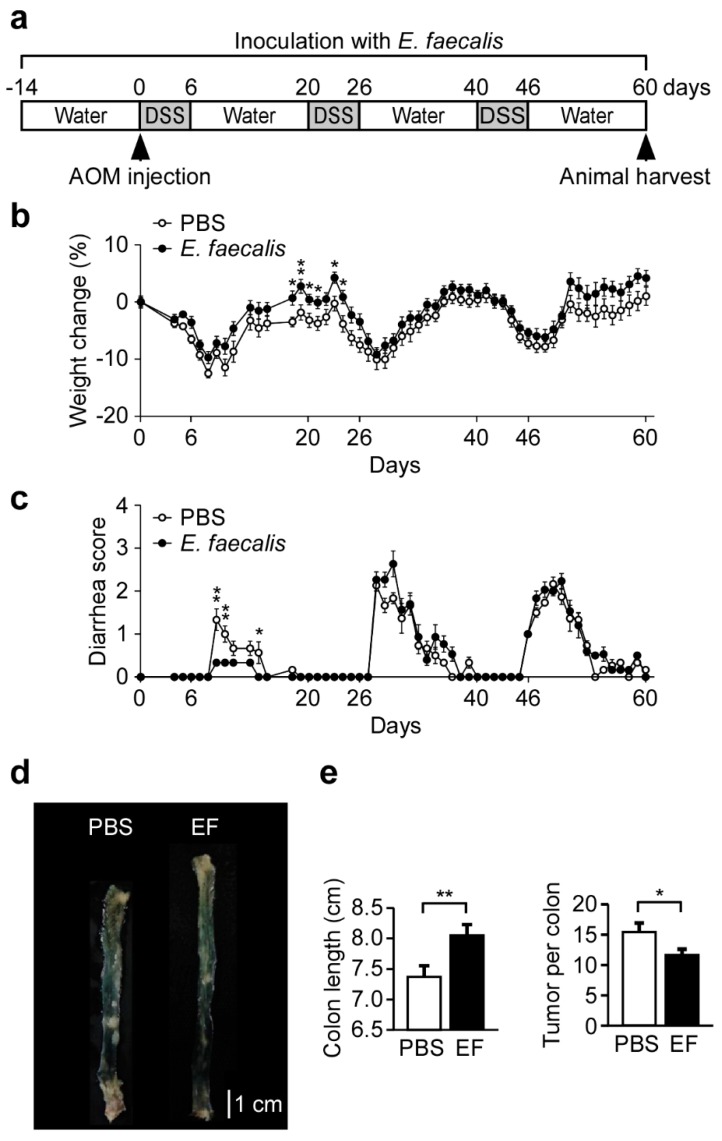
*E. faecalis* pretreatment protects mice from colitis-associated CRC. (**a**) Schematic presentation of the mouse model of colitis-associated CRC. Mice were orally inoculated with *E. faecalis* every day during the course of experiment, starting two weeks before the adminutesistration of DSS. Mice were injected intraperitoneally with AOM prior to beginning the first of three cycles of DSS in the drinking water. One cycle was defined as 6 days of DSS followed by 14 days of water. 60 days after the start of DSS treatment, all mice were sacrificed; *n* = 14 mice for the PBS control and *E. faecalis*-treated groups. (**b**,**c**) Percent weight change (**b**) and diarrhea scores (**c**) were monitored daily in all mice. (**d**) Representative images of colons. (**e**) Colon length and tumor number. Scale bars, 1 cm. *, *p* < 0.05; and **, *p* < 0.01.

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
