# Peer review of "Pretreatment with a Heat-Killed Probiotic Modulates the NLRP3 Inflammasome and Attenuates Colitis-Associated Colorectal Cancer in Mice"

_nutrients, 2019, doi:10.3390/nu11030516_

Round 1

Reviewer 1 Report

The authors have compiled an elegant body of work to demonstrate that pretreatment with heat-killed E. faecalis can inhibit the induction of IL-1β secretion in THP1-derived macrophages stimulated with fecal content and two commensal microbes, P. mirabilis and E. coli. The study further demonstrated E. faecalis attenuates the phagocytosis that is required for full activation of the NLRP3 inflammasome and that E. faecalis can reduce the severity of intestinal inflammation and protect mice from DSS-induced colitis and the formation of CRC.

Some points are listed below:

1.      Have the authors considered mmunohistochemistry data showing staining for NLRP3 or IL-1β complementing the data shown in figure 5. E. faecalis pretreatment protects mice from colitis-associated CRC?

2.      In addition to showing the requirement of the NLRP3 inflammasome for commensal bacteria-induced IL-1β secretion (figure 1), have the authors considered also showing this result with the NLRP3-specific inhibitor, MCC950?

3.      In the methodology, the dilutions, concentrations and catalogue numbers of reagents should be detailed. Eg. In 2.1, Cat # and antibody dilutions for antibodies; In 2.4, the Cat # and concentration of NLRP3 siRNA; In 2.8 The human IL-1β ELISA kit Cat # should be included.

Author Response

Review Report (Reviewer 1)

Comments and Suggestions for Authors

The authors have compiled an elegant body of work to demonstrate that pretreatment with heat-killed E. faecalis can inhibit the induction of IL-1β secretion in THP1-derived macrophages stimulated with fecal content and two commensal microbes, P. mirabilis and E. coli. The study further demonstrated E. faecalis attenuates the phagocytosis that is required for full activation of the NLRP3 inflammasome and that E. faecalis can reduce the severity of intestinal inflammation and protect mice from DSS-induced colitis and the formation of CRC.

Some points are listed below:

1. Have the authors considered mmunohistochemistry data showing staining for NLRP3 or IL-1β complementing the data shown in figure 5. E. faecalis pretreatment protects mice from colitis-associated CRC?

Our response: We appreciate the reviewer’s comments. The previous report by Wang et al. shows that, “In colon tissues, NLRP3 and IL-1β were expressed in both healthy and colitic states as demonstrated by immunohistochemistry staining [20].” (Wang et al. Oncotarget. 2017; 8:32937-32945). In our report, IL-1β was expressed in the colon tissues with DSS treatment in wild-type animals, while in the NLRP3 knockout mice, NLRP3-dependent IL-1β maturation was reduced by E. faecalis pretreatment (Figure 4). In the revised version, we added this reference in the Introduction section (lines 95-96 in the revised version).

2. In addition to showing the requirement of the NLRP3 inflammasome for commensal bacteria-induced IL-1β secretion (figure 1), have the authors considered also showing this result with the NLRP3-specific inhibitor, MCC950?

Our response: We thank the reviewer for reminding us about this inhibitor. Previous reports showed that MCC950 inhibits NLRP3 inflammasome activation and IL-1β secretion in response to Escherichia coli (Schaale et al. Mucosal Immunol. 2016; 9:124-36). In the revised version, we added the following statement in the Discussion secretion. “MCC950 inhibits NLRP3 inflammasome activation and IL-1β secretion in response to Escherichia coli stimulation in macrophages [51].” (lines 420-421 in the revised version).

3. In the methodology, the dilutions, concentrations and catalogue numbers of reagents should be detailed. Eg. In 2.1, Cat # and antibody dilutions for antibodies; In 2.4, the Cat # and concentration of NLRP3 siRNA; In 2.8 The human IL-1β ELISA kit Cat # should be included.

Our response: We added a description of dilutions, concentrations and catalogue numbers of reagents in the methods section of the paragraph 2.1, 2.4, 2.5, 2.7, 2.8 and 2.9 in the revised version:

In 2.1, “PMA (cat # P1585), ATP (cat# A7699), nigericin (cat # N7143), 4',6-diamidino-2-phenylindole (DAPI; cat # D9542) and cytochalasin D (cat # C8273) were purchased from Sigma-Aldrich (St Louis, MO, USA), anti-ASC (cat # SC-22514-R), anti-human caspase-1 (cat # SC-56036), anti-mouse caspase-1 (cat # SC-514), anti-human IL-1β (cat # SC-32294), anti-α-tubulin (cat # SC-32293), anti-GAPDH (cat # SC-32233) and goat anti-rabbit IgG-horseradish peroxidase (HRP; cat # SC-2004) from Santa Cruz Biotechnology (SantaCruz, CA, USA), anti-NLRP3 (cat # AG-20B-0014) from Adipogen (San Diego, CA, USA), anti-mouse IL-1β (cat # AF-401-NA) from R&D Systems Inc. (Minneapolis, MN, USA), sheep anti-mouse IgG-HRP (cat # NA931) from Amersham (Amersham, UK), (5-(and-6)-carboxyfluorescein diacetate, succinimidyl ester (CFSE; cat # C1157) and Alexa Fluor-594 conjugated goat-anti-mouse IgG (H+L) (cat# A-11005) from Invitrogen (Carlsbad, CA, USA) , and fluoresbrite yellow green carboxylate microspheres (1-μm YG beads, cat# 15702) from Polysciences Inc. (Warrington, PA, USA).” (lines 113-124 in the revised version).

In 2.4, “Transfection of the dsRNA duplexes (50 nM) was performed using Lipofectamine 2000 (Invitrogen)” (lines 145-146 in the revised version) and “The reagent used to target NLRP3 (cat# HSS132811-3, Invitrogen) included three 25-bp RNA duplexes” (lines 147-148 in the revised version).

In 2.5, “trichloroacetic acid (TCA; cat # T9159)” (lines 153-154 in the revised version) and “The membranes were incubated with the indicated primary antibodies (1:1000 dilutions of indicated antibody except anti-GAPDH antibody at 1:2000), and then with an HRP-conjugated secondary antibody (1:5000 dilutions of each antibody). The immunoreactive bands were detected using the TOOLS Extreme ECL-HRP Substrate (cat # TU-ECL03; Biotools, Taiwan).” (lines 156-160 in the revised version).

In 2.7, “Cells were stained for visualization of tubulin (1:100 dilution of anti-α-tubulin followed by a 1:1000 dilution of Alexa Fluor-594 conjugated secondary antibody (Invitrogen), red)” (lines 168-170 in the revised version).

In 2.8, “ELISA kit (cat # 88-7261; eBioscience, San Diego, CA, USA)” (lines 177-178 in the revised version).

In 2.9, “Azoxymethane (AOM; 10 mg/kg; cat # A5486; Sigma-Aldrich)” (lines 188 in the revised version).

Detailed modifications in the text:

Introduction:

Addition: NLRP3 and IL-1β were expressed in both healthy and colitic states [20].(lines 95-96 in the revised version).

Materials and Methods:

Modification:PMA, ATP, nigericin, 4',6-diamidino-2-phenylindole (DAPI) and cytochalasin D...” (lines 112-117 in the original version) was changed to “PMA (cat# P1585), ATP (cat# A7699), nigericin (cat# N7143), 4',6-diamidino-2-phenylindole (DAPI; cat# D9542) and cytochalasin D (cat# C8273) were purchased from Sigma-Aldrich (St Louis, Missouri, USA), anti-ASC (cat# SC-22514-R), anti-human caspase-1 (cat# SC-56036), anti-mouse caspase-1 (cat# SC-514), anti-human IL-1β (cat# SC-32294), anti-α-tubulin (cat# SC-32293), anti-GAPDH (cat# SC-32233) and goat anti-rabbit IgG-horseradish peroxidase (HRP; cat# SC-2004) from Santa Cruz Biotechnology (SantaCruz, CA, USA), anti-NLRP3 (cat# AG-20B-0014) from Adipogen (San Diego, CA, USA), anti-mouse IL-1β (cat# AF-401-NA) from R&D Systems Inc. (Minneapolis, MN, USA), sheep anti-mouse IgG-HRP (cat# NA931) from Amersham (Amersham, UK), (5-(and-6)-carboxyfluorescein diacetate, succinimidyl ester(CFSE; cat# C1157) and Alexa Fluor-594 conjugated goat-anti-mouse IgG (H+L) (cat# A-11005) from Invitrogen (Carlsbad, CA, USA) , and fluoresbrite yellow green carboxylate microspheres (1-μm YG beads, cat# 15702) from Polysciences Inc. (Warrington, PA, USA).” (lines 113-124 in the revised version).

Addition: “(50 nM)” (lines 145 in the revised version).

Addition: “(cat# HSS132811-3)” (lines 147 in the revised version).

Addition: “(TCA; cat# T9159)” (lines 154 in the revised version).

Addition: “(1:1000 dilutions of indicated antibody except anti-GAPDH antibody at 1:2000)” (lines 157 in the revised version).

Addition: “(1:5000 dilutions of each antibody)” (lines 158 in the revised version).

Modification: “(Biotools)” (lines 153 in the original version) was changed to “(cat# TU-ECL03; Biotools, Taiwan)” (lines 159-160 in the revised version).

Modification: “tubulin (anti-α-tubulin, blue)” (lines 161-162 in the original version) was changed to “tubulin (1:100 dilution of anti-α-tubulin followed by a 1:1000 dilution of Alexa Fluor-594 conjugated secondary antibody (Invitrogen), red)” (lines 168-170 in the revised version).

Modification: “(eBioscience)” (lines 170 in the original version) was changed to “(cat# 88-7261; eBioscience, San Diego, CA, USA)” (lines 177 in the revised version).

Modification: “AOM (10 mg/kg)” (lines 180 in the original version) was changed to “Azoxymethane (AOM; 10 mg/kg; cat# A5486; Sigma-Aldrich)” (lines 188 in the revised version).

Addition: “Scoring for severity of diarrhea was performed as described previously [30] with a slight modification. Briefly, diarrhea scores were determined as follows: 0 points for well-formed stools, 1 point for soft but formed stools, 2 points for pasty and semiformed stools that did not adhere to the anus (mild diarrhea), 3 points for slimy stool (moderate diarrhea). 4 points for liquid stools that did adhere to the anus (severe diarrhea).” (lines 193-197 in the revised version).

Discussion:

Addition: MCC950 can inhibit NLRP3 inflammasome activation and IL-1β secretion in response to Escherichia coli stimulation in macrophages [51].(lines 420-421 in the revised version).

References:

Addition: 20. Wang, L.; Yu, Z.; Wei, C.; Zhang, L.; Song, H.; Chen, B.; Yang, Q. Huaier aqueous extract protects against dextran sulfate sodium-induced experimental colitis in mice by inhibiting nlrp3 inflammasome activation. Oncotarget 2017, 8, 32937-32945.

Addition: 51.   Schaale, K.; Peters, K.M.; Murthy, A.M.; Fritzsche, A.K.; Phan, M.D.; Totsika, M.; Robertson, A.A.; Nichols, K.B.; Cooper, M.A.; Stacey, K.J., et al. Strain- and host species-specific inflammasome activation, il-1beta release, and cell death in macrophages infected with uropathogenic escherichia coli. Mucosal Immunol 2016, 9, 124-136.

Reviewer 2 Report

In the paper entitled “Pretreatment with a heat-killed probiotic modulates the NLRP3 inflammasome and prevents colitis-associated colorectal cancer in mice” by Dr. I-Che Chung et al, the authors explored the possibility that the treatment of cells macrophage-like cells with heat-killed Enterococcus faecalis can have a beneficial effect on inflammation thus reducing the risk of colitis associated colorectal cancer (CRC). The results they found confirmed their hypothesis, showing a benefit in the use of the probiotic in terms of lower inflammation as well as in decreasing the occurrence of CRC. The manuscript is well structure, well written and easy to follow. There are, however, some comments:

1)     In the abstract (line 49) and later on in the text (e.g. line 82), the authors wrote the acronym DSS. Please write the acronym meaning at the first time of encounter, both in the abstract and in the text.

2)     In the abstract, line 47, please correct the word “experiemtns” to “experiments”.

3)     On line 180, please explain the acronym AOM.

4)     In the methods section, line 151 please indicate the dilution of primary antibodies.

5)     Which post-hoc test did the authors use after ANOVA? It is not explained in the statistical analysis section.

6)     In the results section (page 5, paragraph 3.1), the authors said that the induction of IL-1beta is inhibited by NLRP3 knockdown. However, I think that the proper term should be reduced or at least partially inhibited by the NLRP3 knockdown, since from figure 1 it seems that the knockdown itself was not almost 100% efficient.

7)     In the results, why the authors did search active caspase-1 in the cell supernatants and not in the cell lysates? Caspase -1 should be an intracellular protein and its presence should be checked in the cell lysates.

8)     In the results of paragraph 3.2 and In Figure 2, why did the authors do different experiments (and different conditions) from panel d and c and did not do a dose-dependency of E. faecalis on Ec and Pm strains?

9)     In the Figure 2 legend, the authors claim that the panel d is referred to heat-inactivated and UV-irradiated fecal content, and that panel e regards the experiments on Ec and Pm. However, the results regarding the heat-inactivated and UV-irradiated fecal contents are not present in this version of the manuscript, and the panels in Figure 2 start with “a” and end with “d”. Please correct.

10)  In the paragraph 3.3 the authors said that the pretreatment of cells with inactivated E. faecalis decrease bacterial phagocytosis. I’m not fully convinced of this result since immunofluorescence pictures of figure 3 poorly represent the event, with the exception of the one in panel a. If these are just representative fields, it would be appropriate to choose some showing more evidently the effect.

11)  Regarding the experiments in paragraph 3.3, the authors just stop after observing the inhibition of phagocytosis. For instance, it would be of interest if they observe the activation/interference with Rho GTPase since this family of proteins are highly important for the control of phagocytosis and could improve the mechanistic view of the manuscript.

12)  On line 273, within the firs brackets the anti-tubulin is marked as blue as the DAPI. Please place the right color for tubulin.

13)  In the results of the paragraph 3.5, how was the diarrhea score calculated?

14)  Although the authors claim that E. faecalis treatment on animal may exert preventive effects against CRC, I’m not fully convinced. In fact, the animals treated with EF, although with an improved “disease status” do develop CRC. So the effect may be protective or EF treatment can attenuate the appearance of CRC due to the modest decrease in tumors per colon.

Author Response

Review Report (Reviewer 2)

Comments and Suggestions for Authors

In the paper entitled “Pretreatment with a heat-killed probiotic modulates the NLRP3 inflammasome and prevents colitis-associated colorectal cancer in mice” by Dr. I-Che Chung et al, the authors explored the possibility that the treatment of cells macrophage-like cells with heat-killed Enterococcus faecalis can have a beneficial effect on inflammation thus reducing the risk of colitis associated colorectal cancer (CRC). The results they found confirmed their hypothesis, showing a benefit in the use of the probiotic in terms of lower inflammation as well as in decreasing the occurrence of CRC. The manuscript is well structure, well written and easy to follow. There are, however, some comments:

1) In the abstract (lines 49) and later on in the text (e.g. lines 82), the authors wrote the acronym DSS. Please write the acronym meaning at the first time of encounter, both in the abstract and in the text.

Our response: We now define DSS the first time that the term is used in each section (lines 48, 82 and 97 in the revised version).

2) In the abstract, lines 47, please correct the word “experiemtns” to “experiments”.

Our response: We corrected the mis-spelling (lines 47 in the revised version).

3) On lines 180, please explain the acronym AOM.

Our response: We now define it in the text (lines 188 in the revised version).

4) In the methods section, lines 151 please indicate the dilution of primary antibodies.

Our response: We added a description of the dilutions of primary antibodies in the methods section in paragraph 2.5 in the revised version as follows: “The membranes were incubated with the indicated primary antibodies (1:1000 dilutions of indicated antibody except anti-GAPDH antibody at 1:2000)” (lines 156-157 in the revised version).

5) Which post-hoc test did the authors use after ANOVA? It is not explained in the statistical analysis section.

Our response: We performed Bonferroni post hoc test after ANOVA. In the revised version, we modified the Materials and Methods to make this clear. “Data from in vitro experiments and tumor growth in the mouse model were analyzed with the ANOVA test followed by Bonferroni post hoc test as indicated or Student’s t test in SPSS as described previously [30].(lines 200-202 in the revised version).

6) In the results section (page 5, paragraph 3.1), the authors said that the induction of IL-1beta is inhibited by NLRP3 knockdown. However, I think that the proper term should be reduced or at least partially inhibited by the NLRP3 knockdown, since from figure 1 it seems that the knockdown itself was not almost 100% efficient.

Our response: We changed the term “inhibited” to “reduced” in the following statement. Similarly, active caspase-1, mature IL-1β and secreted IL-1β were induced by infection with either E. coli or P. mirabilis, and this induction was reduced by NLRP3 knockdown (Figure 1b).(lines 212-213 in the revised version).

7) In the results, why the authors did search active caspase-1 in the cell supernatants and not in the cell lysates? Caspase -1 should be an intracellular protein and its presence should be checked in the cell lysates.

Our response: The secretion of active caspase-1 along with mature IL-1β into the supernatant was shown in the first description of the inflammasome (Martinon et al. Mol Cell. 2002; 2:417-26). Although pro-caspase-1 (45 kDa) can be detected in the cell lysates, active caspase-1 (10 kDa) is secreted by the cells (Ayala et al. J Immunol. 1994;153:2592-9). Therefore, we measured pro-caspase-1 (45 kDa) and active caspase-1 (10 kDa) in cell lysates and cell supernatants, respectively.

8) In the results of paragraph 3.2 and In Figure 2, why did the authors do different experiments (and different conditions) from panel d and c and did not do a dose-dependency of E. faecalis on Ec and Pm strains?

Our response: The suppressive effect of E. faecalis on fecal content-induced IL-1β secretion appears to be dose-dependent (Figure 2c) and achieved maximum effect at a E. faecalis dose of 2 mg/ml. Therefore, we used 2 mg/ml E. faecalis to evaluate its suppressive effect on Ec and Pm strains-induced IL-1β secretion (Figure 2d).

9) In the Figure 2 legend, the authors claim that the panel d is referred to heat-inactivated and UV-irradiated fecal content, and that panel e regards the experiments on Ec and Pm. However, the results regarding the heat-inactivated and UV-irradiated fecal contents are not present in this version of the manuscript, and the panels in Figure 2 start with “a” and end with “d”. Please correct.

Our response: We corrected the mistakes and modified the description as follows in the Figure 2 legend: “(c-d) ELISA of IL-1β in the SN of THP-1-derived macrophages pretreated with E. faecalis at the dose as indicated (c) or 2 mg/ml (d) for 24 h and then infected for 3 h with FC (c), or Ec or Pm at 20, 50 or 100 MOI (d).(lines 249-251 in the revised version).

10) In the paragraph 3.3 the authors said that the pretreatment of cells with inactivated E. faecalis decrease bacterial phagocytosis. I’m not fully convinced of this result since immunofluorescence pictures of figure 3 poorly represent the event, with the exception of the one in panel a. If these are just representative fields, it would be appropriate to choose some showing more evidently the effect.

Our response: We replaced the representative image of Proteus mirabilis-challenged cells with E. faecalis pretreatment (Figure 3b, bottom right panel). In Figure 3b and 3c, we added closed arrowheads and open arrowheads to denote the phagocytosed (intracellular) and unphagocytosed (extracellular) particles (fecal content, E. coli, or P. mirabilis), respectively. In addition, we added supplementary data to show more representative fields (Figure S1 in the revised version). In the revised version, we also modified the Results section: “Similarly, E. faecalis also significantly inhibited the phagocytosis of fecal content, E. coli and P. mirabilis (Figure 3b and Figure S1).” (lines 268-269 in the revised version).

Figure 3. Phagocytosis, which is required for bacteria-induced inflammasome activation, is inhibited by E. faecalis. (a and b) THP-1-derived macrophages were pretreated with E. faecalis (Ef, 2 mg/ml) for 24 h, incubated for 3 h with (a) beads, (b) FC, Ec or Pm, and assessed for phagocytosis. Scale bars, 20 μm. Phagocytosis was quantified with an IN Cell Analyzer. (c) Assessment of phagocytosis in THP-1-derived macrophages pretreated with cytochalasin D for 1 h and then incubated for 3 h with FC. Scale bars, 10 μm. Cells were stained for visualization of tubulin (anti-a-tubulin, red), beads, FC, Ec or Pm (CFSE, green) and nuclei (DAPI, blue). (d) ELISA of IL-1β in the SN of THP-1-derived macrophages pretreated with cytochalasin D for 1 h and then infected for 3 h with FC, Ec or Pm. *, P < 0.05; and **, P < 0.01. Arrow, phagocytosed E. faecalis. Closed arrowhead, phagocytosed FC, Ec or Pm. Open arrowhead, unphagocytosed FC, Ec or Pm. These data are representative of three independent experiments.

Figure S1. Bacterial phagocytosis is inhibited by E. faecalis pretreatment. THP-1-derived macrophages were pretreated with E. faecalis (2 mg/ml) for 24 h, incubated for 3 h with fecal content (FC), E. coli (Ec) or P. mirabilis (Pm), and assessed for phagocytosis. Cells were stained for visualization of tubulin (anti-a-tubulin, red), FC, Ec or Pm (CFSE, green) and nuclei (DAPI, blue). Square, the field shown in the Figure 3b. Arrowhead, the cells with phagocytosed FC, Ec or Pm. Scale bars, 20 μm.

11)  Regarding the experiments in paragraph 3.3, the authors just stop after observing the inhibition of phagocytosis. For instance, it would be of interest if they observe the activation/interference with Rho GTPase since this family of proteins are highly important for the control of phagocytosis and could improve the mechanistic view of the manuscript.

Our response: Rho GTPase proteins are in fact important control of phagocytosis. Beyond showing the dependence on phagocytosis, we observed that phagocytosis of many E. faecalis caused by E. faecalis pretreatment resulted in inhibition of phagocytosis of polystyrene latex beads (Figure 3a), fecal content, E. coli and P. mirabilis (Figure 3b). The results suggest that the capacity of macrophages to phagocytose may be exhausted by pretreatment with E. faecalis. In the revised version, we added the following statement in the Results section. “As shown in Figure 3a, pretreatment of E. faecalis caused clearly intracellular, phagocytosed E. faecalis(lines 265-266 in the revised version) and “The phagocytosed E. faecalis appear to exhaust the capacity of phagocytosis in macrophages.” (lines 267-268 in the revised version).

12)  On lines 273, within the firs brackets the anti-tubulin is marked as blue as the DAPI. Please place the right color for tubulin.

Our response: We corrected the mistake (lines 170 and lines 285 in the revised version).

13)  In the results of the paragraph 3.5, how was the diarrhea score calculated?

Our response: We added a sentence to calculate the diarrhea score in the Material and Methods section (paragraph 2.9) in the revised version: “Scoring for severity of diarrhea was performed as described previously [30] with a slight modification. Briefly, diarrhea scores were determined as follows: 0 points for well-formed stools, 1 point for soft but formed stools, 2 points for pasty and semiformed stools that did not adhere to the anus (mild diarrhea), 3 points for slimy stool (moderate diarrhea), 4 points for liquid stools that did adhere to the anus (severe diarrhea).” (lines 193-197 in the revised version).

14)  Although the authors claim that E. faecalis treatment on animal may exert preventive effects against CRC, I’m not fully convinced. In fact, the animals treated with EF, although with an improved “disease status” do develop CRC. So the effect may be protective or EF treatment can attenuate the appearance of CRC due to the modest decrease in tumors per colon.

Our response: We agree with the reviewer’s comments. In the revised version, we replaced “prevent” with “attenuate” (lines 3, 51, 337, 365 and 371 in the revised version) and “preventive” with “protective” (lines 349 in the revised version).

Detailed modifications in the text:

Title:

Modification: “prevents” (lines 3 in the original version) was changed to attenuates (lines 3 in the revised version).

Abstract:

Modification: “experiemtns” (lines 47 in the original version) was changed to experiments (lines 47 in the revised version).

Modification: “DSS” (lines 48 in the original version) was changed to dextran sodium sulfate (DSS) (lines 48 in the revised version).

Modification: “prevention” (lines 51 in the original version) was changed to attenuation (lines 51 in the revised version).

Introduction:

Modification: “DSS” (lines 82 in the original version) was changed to dextran sodium sulfate (DSS) (lines 82 in the revised version).

Modification: “dextran sodium sulfate (DSS)” (lines 96 in the original version) was changed to DSS (lines 97 in the revised version).

Materials and Methods:

Modification: “All statistical analyses were performed using the SPSS 13.0 statistical software package (SPSS). Data from in vitro experiments and tumor growth in the mouse model were analyzed with the ANOVA test followed by post hoc test as indicated or Student’s t test.” (lines 188-190 in the original version) was changed to “Data from in vitro experiments and tumor growth in the mouse model were analyzed with the ANOVA test followed by Bonferroni post hoc test as indicated or Student’s t test in SPSS as described previously [30].(lines 200-202 in the revised version).

Results:

Modification: “Similarly, the amounts of active caspase-1, mature IL-1β and secreted IL-1β were induced by infection with either E. coli or Pmirabilis, and this induction was inhibited by NLRP3 knockdown (Figure 1b).(lines 201-202 in the original version) was changed to “Similarly, active caspase-1, mature IL-1β and secreted IL-1β were induced by infection with either E. coli or P. mirabilis, and this induction was reduced by NLRP3 knockdown (Figure 1b).(lines 212-213 in the revised version).

Modification: “Similarly, E. faecalis also significantly inhibited the phagocytosis of fecal content, E. coli and P. mirabilis (Figure 3b).(lines 256-257 in in the original version)was changed to “Similarly, E. faecalis also significantly inhibited the phagocytosis of fecal content, E. coli and P. mirabilis (Figure 3b and Figure S1). (lines 268-269 in in the revised version).

Addition: “As shown in Figure 3a, pretreatment of E. faecalis caused clearly intracellular, phagocytosed E. faecalis(lines 265-266 in the revised version).

Addition: “The phagocytosed E. faecalis appear to exhaust the capacity of phagocytosis in macrophages. (lines 267-268 in the revised version).

Modification: “prevent” (lines 325 in the original version) was changed to “attenuate” (lines 337 in the revised version).

Modification: “preventive” (lines 337 in the original version) was changed to “protective” (lines 349 in the revised version).

Discussion:

Modification: “prevent” (lines 354 in the original version) was changed to “attenuate” (lines 365 in the revised version).

Modification: “prevention” (lines 359 in the original version) was changed to “attenuation” (lines 371 in the revised version).

References:

Addition: 30. Chung, I.C.; Yuan, S.N.; OuYang, C.N.; Lin, H.C.; Huang, K.Y.; Chen, Y.J.; Chung, A.K.; Chu, C.L.; Ojcius, D.M.; Chang, Y.S., et al. Src-family kinase-cbl axis negatively regulates nlrp3 inflammasome activation. Cell Death Dis 2018, 9, 1109.

Addition: 31.   Siegmund, B.; Lehr, H.A.; Fantuzzi, G.; Dinarello, C.A. Il-1 beta -converting enzyme (caspase-1) in intestinal inflammation. Proceedings of the National Academy of Sciences of the United States of America 2001, 98, 13249-13254.

Figure legends:

Modification: Figure 2. “(c-e) ELISA of IL-1β in the SN of THP-1-derived macrophages…” (lines 238-240 in the original version) was changed to (c-d) ELISA of IL-1β in the SN of THP-1-derived macrophages pretreated with E. faecalis at the dose as indicated (c) or 2 mg/ml (d) for 24 h and then infected for 3 h with FC (c), or Ec or Pm at 20, 50 or 100 MOI (d). (lines 249-251 in the revised version).

Modification: Figure 3. “blue” (lines 273 in the original version) was changed to red (lines 285 in the revised version).

Addition: “Closed arrowhead, phagocytosed FC, Ec or Pm. Open arrowhead, unphagocytosed FC, Ec or Pm. (lines 287-288 in the revised version).

Supplementary data:

Addition: Figure S1.

Figure S1. Bacterial phagocytosis is inhibited by E. faecalis pretreatment. THP-1-derived macrophages were pretreated with E. faecalis (2 mg/ml) for 24 h, incubated for 3 h with fecal content (FC), E. coli (Ec) or P. mirabilis (Pm), and assessed for phagocytosis. Cells were stained for visualization of tubulin (anti-a-tubulin, red), FC, Ec or Pm (CFSE, green) and nuclei (DAPI, blue). Square, the field shown in the Figure 3b. Arrowhead, the cells with phagocytosed FC, Ec or Pm. Scale bars, 20 μm.

Round 2

Reviewer 2 Report

I feel that the authors correctly replied to the concerns I made and the manuscript improved.